Corrected: Author Correction

# Proteome evolution under non-substitutable resource limitation

Manu Tamminen[1,2], Alexander Betz[3], Aaron Louis Pereira[2], Marco Thali[2], Blake Matthews[4] Marc J.-F. Suter[3] & Anita Narwani[2]

Resource limitation is a major driver of the ecological and evolutionary dynamics of organisms. Short-term responses to resource limitation include plastic changes in molecular phenotypes including protein expression. Yet little is known about the evolution of the molecular phenotype under longer-term resource limitation. Here, we combine experimental evolution of the green alga *Chlamydomonas reinhardtii* under multiple different non-substitutable resource limitation regimes with proteomic measurements to investigate evolutionary adaptation of the molecular phenotype. We demonstrate convergent proteomic evolution of core metabolic functions, including the Calvin-Benson cycle and gluconeogenesis, across different resource limitation environments. We do not observe proteomic changes consistent with optimized uptake of particular limiting resources. Instead, we report that adaptation proceeds in similar directions under different types of non-substitutable resource limitation. This largely convergent evolution of the expression of core metabolic proteins is associated with an improvement in the resource assimilation efficiency of nitrogen and phosphorus into biomass.

[1] Department of Biology, University of Turku, Natura, Yliopistonmäki, 20014 Turku, Finland. [2] Department of Aquatic Ecology, Eawag, Überlandstrasse 133, 8600 Dübendorf, Switzerland. [3] Department of Environmental Toxicology, Eawag, Überlandstrasse 133, 8600 Dübendorf, Switzerland. [4] Department of Aquatic Ecology, Eawag, 79 Seestrasse, 6047 Kastanienbaum, Switzerland. Correspondence and requests for materials should be addressed to A.N. (email: anita.narwani@eawag.ch)

The ability to survive on and compete for limited resources is an important factor governing the abundance and distribution of all organisms[1]. Resource limitation can determine the persistence and dynamics of populations, and the outcome of competitive interactions among species. Phytoplankton, like all photo-autotrophs, compete for non-substitutable and limiting inorganic resources, such as light, nitrogen, and phosphorus, among others[2]. Population-level responses of phytoplankton to resource limitation vary widely across the tree of life[3–5], and recent evidence suggests that these responses may evolve rapidly[6]. In this paper, we investigate the molecular basis of adaptation to limiting resources in phytoplankton.

Quantifying metabolic shifts induced by resource limitation can provide important insights into the plastic and evolutionary adaption of phytoplankton to environmental change. Whole-genome sequencing of model organisms, such as *Chlamydomonas reinhardtii*, and high-throughput molecular phenotyping methods, including transcriptomics, proteomics, and metabolomics, have been used in studies focusing on plastic changes in the molecular phenotype induced by short-term exposure to resource limitation[7–9]. These studies have demonstrated that phytoplankton may show significant phenotypic restructuring at the metabolic and molecular level, which may enable species to cope with fluctuations and short-term shortages in resource availability. However, virtually nothing is known about evolutionary adaptation occurring at the level of the molecular phenotype in response to longer-term exposure to non-substitutable resource limitation (but see refs. [10,11]). We aim here to address this gap using a powerful combination of experimental evolution and proteomic phenotyping.

Phenotypic evolution in response to different nutrient limitation scenarios can be either convergent or divergent. One possibility is that competition for limiting resources causes phenotypic divergence[12–15]. For example, variation in the identity of resource limitation in a heterogeneous environment may select for unique phenotypes which are each specialized on efficiently consuming and converting different limiting resources, according to availability[11,16–19]. An alternative outcome could be convergent evolution if similar phenotypes are repeatedly selected from independent origins due to similar types of environmental selection (e.g., limitation by the same resource)[20–22].

Here, we propose that selection under different types of resource limitation may also select for convergent trait evolution, particularly when the resources are non-substitutable, and therefore essential for growth. In the case of non-substitutable resource limitation, blockage of the cell cycle, growth, and therefore fitness, cannot be overcome by adaptive specialization on other, more available resources. In this case, adaptation can only result from (a) increased capacity to acquire the non-substitutable limiting resource or (b) increased metabolic efficiency given a fixed ability to acquire the non-substitutable limiting resource. In the first case, phenotypic adaptation may be specific to the identity of the limiting resource (i.e., divergent in the case of different limiting resources), but in the second case, phenotypic adaptation is likely to involve changes in core metabolism that are similar across a variety of limiting resources (i.e., convergent), especially when phenotypic changes are strongly constrained by metabolism.

In phytoplankton, there are several well-known core metabolic functions that are necessary for reproduction, including photosynthesis, respiration, carbon metabolism, transcription, translation, and protein synthesis. Short-term metabolic changes in phytoplankton in response to various types of resource-induced stress, including nitrogen or phosphorus limitation, involve shifts in core carbon metabolism including photosynthesis, the Calvin–Benson cycle, glycolysis/gluconeogenesis, and fatty acid synthesis, as well as shifts in the expression of proteins used to protect from heat-shock or photo-oxidative damage[7,8,10,23–25]. While not universal, resource-induced stress tends to lead to a reduction in photosynthesis and the Calvin–Benson cycle, and a concurrent increase in starch production or fatty acid biosynthesis[7,23,25]. More generally, these responses represent tweaks of core metabolism that allow cells to shift resource allocation from growth to storage, given the physiochemical constraints on the cell cycle imposed by the resource limitation. Nevertheless, resource limitation also induces molecular phenotypic changes that are unique and targeted to the specific resource and are functionally targeted at improved resource acquisition. For example, both nitrogen[25] and phosphorus limitation[8] result in the upregulation of the respective nutrient transporters and organic nutrient assimilation pathways, and low light results in the upregulation of photosynthesis proteins[24]. There may therefore be both convergent and divergent phenotypic adaptations occurring simultaneously in response to selection under unique but non-substitutable resource limitation, with convergent responses occurring in core metabolism and divergent responses occurring in resource acquisition strategies.

Here we used experimental evolution under different nutrient limitation scenarios to investigate the metabolic adaptation of the model green phytoplankton species *Chlamydomonas reinhardtii*. We aimed to address three basic questions: (1) Does protein expression evolve in response to limitation under non-substitutable resource limitation? (2) If so, is proteomic evolution generally convergent or divergent in response to different types of non-substitutable resource limitation? (3) Are convergent responses overrepresented by core metabolic functions and divergent responses overrepresented by resource acquisition functions? We investigated the model alga, *C. reinhardtii*, because it is known to evolve rapidly and has exceptional genomic resources[26–30]. We grew *C. reinhardtii* either under light, nitrogen or phosphorus limitation, as well as in a biotically-spent medium (i.e., the medium had been used to grow other algae prior to being fed to the evolving *C. reinhardtii*). We used a biotically-spent medium to mimic the influence that a biodiverse community of phytoplankton may have on the availability of all dissolved resources: multiple resources being depleted simultaneously according to the requirements of biotic competitors. We also introduced a salt-stress selection environment, and a crossed salt-stress by biotically-spent medium environment, because previous work has suggested that stress may accelerate adaptive evolution[31] and thus accelerate metabolic adaptations over the course of the period of the experimental evolution. Furthermore, salt-stress provides an outlier selection treatment which enables a comparison of adaptation under resource limitation relative to that under another type of environmental stress. We report adaptation which proceeds in similar directions under different types of non-substitutable resource limitation. This convergent evolution is associated with an improvement in the resource assimilation efficiency of nitrogen and phosphorus into biomass.

## Results

**Evidence of proteomic evolution under resource limitation.** Selection under resource limitation (as outlined in Supplementary Figure 1) resulted in significant evolution of *Chlamydomonas reinhardtii* proteomes, as indicated by the redundancy analysis (RDA) and permutational ANOVA ($p < 0.001$; Fig. 1). RDA was performed using all 3347 protein expression levels as response variables and populations and treatments as explanatory variables, providing 51 independent observations. The first two RDA axes explain 74% of the total variation. We treated the

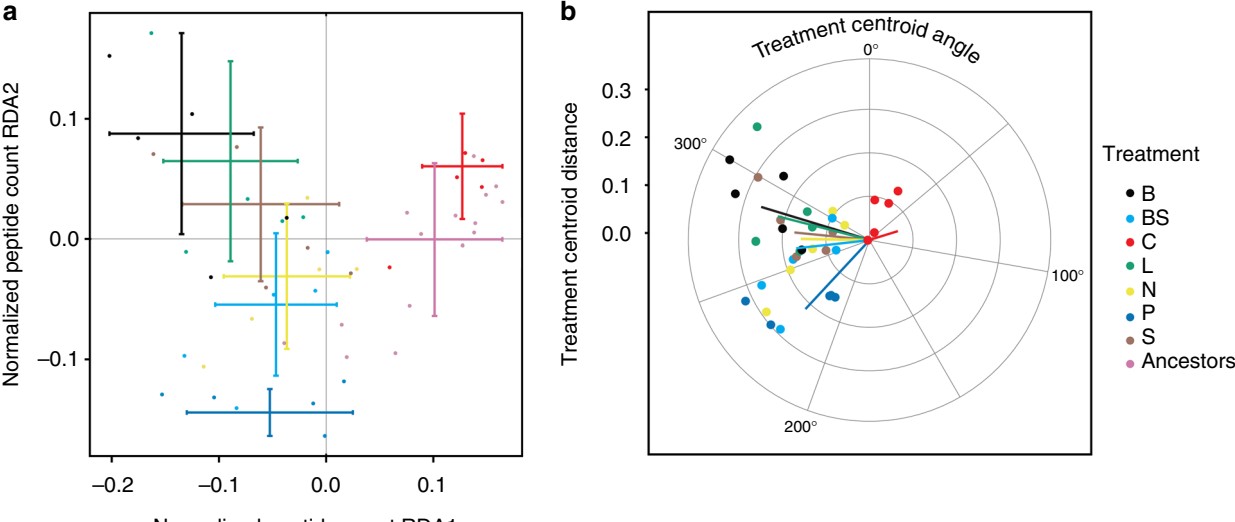

**Fig. 1** RDA analysis of protein expression levels explained by treatments and conditioned by population. Protein expression levels include 3347 response variables and consist of 51 independent observations. **a** The selection treatments cause significant differences in the protein expression levels (redundancy analysis, RDA; permutational ANOVA; $p < 0.001$; error bars represent standard errors). The first two RDA axes explain 74% of the total variation (which totals at 0.00586 units). Ancestral and control treatments are separated from the other treatments by Axis 1 (ANOVA; TukeyHSD post hoc test $p < 0.01$) and the low-phosphate treatment is separated from ancestors, controls, biotic, low-light, and high-salt treatments by Axis 2 (ANOVA; TukeyHSD post hoc test $p < 0.01$). **b** The angles and distances of each treatment from the ancestral mean on the RDA plane. All treatments exhibit convergent trajectories with the significant exception of the control ($p < 0.01$; TukeyHSD post hoc test)

populations as biological replicates, as there were no differences in the proteome expression among the populations when placed in a common garden after the selection treatments (RDA analysis with permutational ANOVA; $p = 0.762$) (Supplementary Figure 2).

We quantified 3347 proteins and found that 1304 displayed significant differential regulation relative to the ancestral average (permutation test $p < 0.001$; Supplementary Figure 4), and 351 showed significant differential regulation relative to the control selection lines (Fig. 2). We found little evidence for strong adaptation to life in chemostat, as expression between the control treatments and ancestors was not significantly different (ANOVA; TukeyHSD post hoc test of RDA axis 1 $p = 0.99$; RDA axis 2 $p = 0.75$) (Fig. 1, Supplementary Figure 3). However, we found significant protein expression differences among treatments — ancestors and control treatment were significantly different from all other treatments on RDA Axis 1 (ANOVA; $p < 0.01$) (Fig. 1, Supplementary Figure 3). The low-phosphate treatment was significantly different from the ancestors and the control, biotic, low-light, and high-salt treatments on RDA Axis 2 (ANOVA; $p < 0.01$) (Fig. 1, Supplementary Figure 3). The salt treatment, included in this experiment as an outlier treatment, representative of stress in general, was not significantly different from any of the resource limitation treatments.

**Convergence versus divergence in proteomic evolution.** We identified support for convergent proteomic evolution in six out of seven treatments by measuring the centroid, direction, and distance of each treatment relative to the ancestral mean. The selection treatments significantly affected the direction of evolution relative to the ancestors on the plane of the two first RDA axes (ANOVA; $p < 0.001$) (Fig. 1b). The control treatments were separate from the other treatments in the direction of their evolution (TukeyHSD post hoc test; $p < 0.05$) (Supplementary Figure 4). The evolutionary distance, as quantified by the distance from the ancestral mean on the plane of the first two RDA axes,

was not significantly different among the treatments (ANOVA; $p = 0.297$) (Fig. 1b and Supplementary Figure 4).

In order to divide proteins into those showing convergent and divergent responses to selection across treatments, we first identified the proteins whose expression was significantly different relative to the control treatment (Dunnett test $p < 0.05$; $n = 351$) and then divided them into convergent (Friedman test $p >= 0.1$; $n = 232$) and divergent (Friedman test $p < 0.1$; $n = 70$) responses, based on whether or not treatments had significantly different impacts on protein expression. Both convergent and divergent proteins are non-uniformly distributed across different genomic and chromosomal loci, with certain chromosomes showing little variation (such as chromosomes 14 and 15) and other chromosomes showing a high degree of variation (such as chromosomes 6 and 12) (chi-square test $p < 0.001$) (Fig. 2b). We found evidence for extensive convergent proteomic evolution in response to the different selection treatments.

**Functional classification of proteins responding to selection.** Among proteins with a convergent response (Friedman test $p >= 0.1$), 89 proteins generally responded positively across all selection treatments (Group 1 in Fig. 3), 178 generally responded negatively (Group 2). Among proteins with a divergent response (Friedman test $p < 0.1$), 43 proteins generally responded positively (Group 3) and 44 generally negatively (Group 4) (Fig. 3). In addition, we detected 14 chloroplast-encoded genes, 11 of which exhibited a convergent, and 3 a divergent response (Supplementary Figure 5). Several proteins across the divergent and convergent responses have control treatment values, which are significantly different from the ancestral values (t test $p < 0.01$; marked with asterisks; convergent $n = 34$; divergent $n = 11$). We consider these proteins examples of adaptation to chemostat conditions, which are different from the ancestral growth environment.

We performed enrichment analysis for proteins within Groups 1, 2, 3, and 4 using DAVID Functional Annotation Tool. Positively expressed proteins (Groups 1 and 3) in both

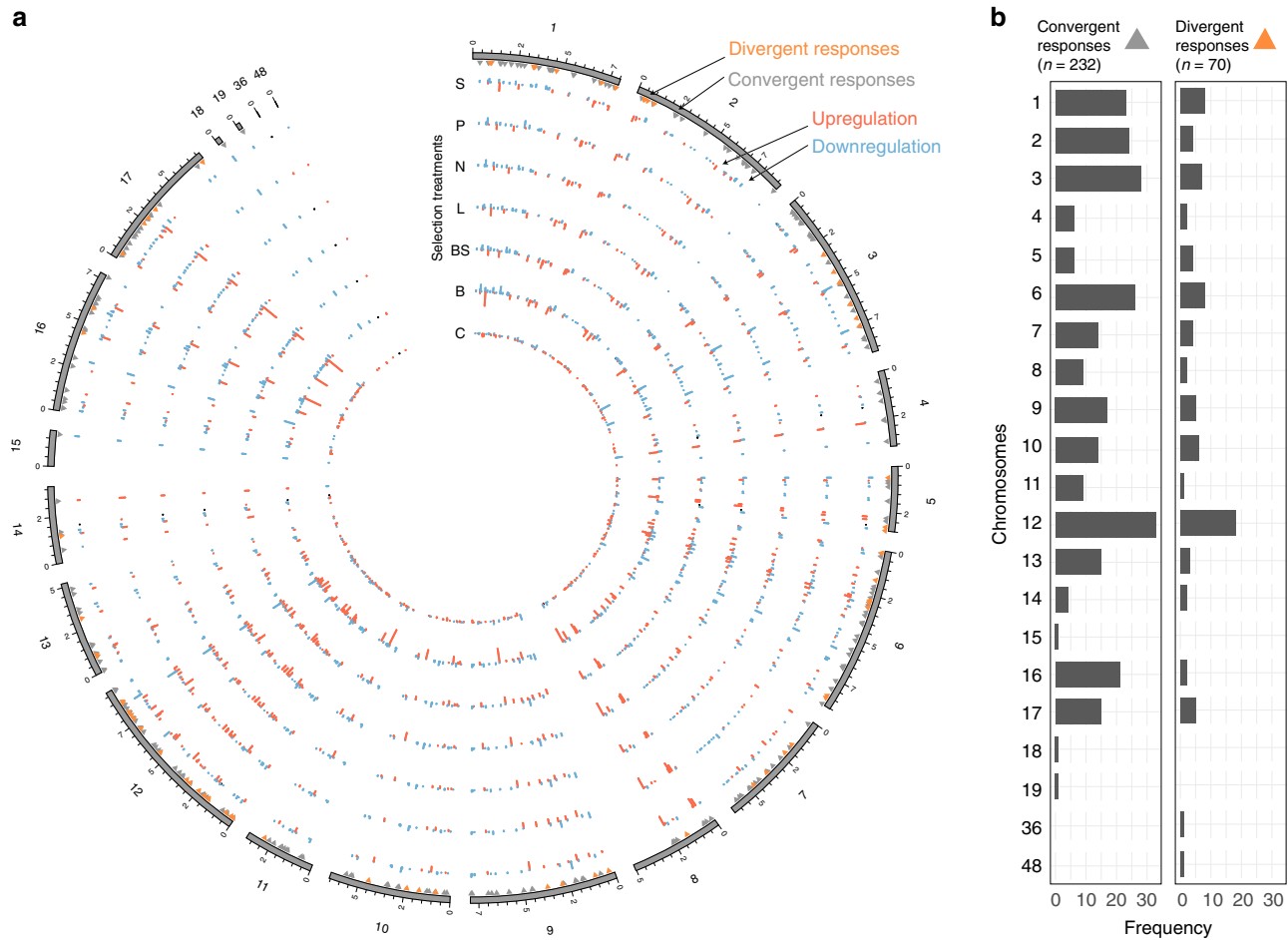

**Fig. 2** Chromosomal loci of the differentially expressed proteins. **a** Proteins where the expression of at least one treatment is significantly different from the ancestors ($t$ test $p < 0.01$; $n = 1304$; 39% of the total detected proteins). **b** Among these, 351 proteins exhibit a differential response from the control (Dunnett test $p < 0.05$). These differentially expressed proteins classify into divergent (Friedman test $p < 0.1$; $n = 70$) and convergent responses (Friedman test $p > = 0.1$; $n = 232$). Both convergent and divergent responses are non-randomly distributed across the chromosomes (chi-square test; $p < 0.0001$)

convergent and divergent protein groups consist of ribosome- and photosynthesis-related proteins (DAVID Functional Annotation Clustering; $p < 0.01$). Negatively expressed proteins within the convergent group (Group 2) consists of carbon metabolism-related proteins, ATP binding proteins, pyridoxal phosphate-dependent transferases, fatty acid degradation-related proteins, and plant-type vacuolar proteins (DAVID Functional Annotation Clustering; $p < 0.01$). Negatively expressed proteins within the divergent group (Group 4) do not contain any significantly enriched functions. Unsurprisingly, chloroplast-encoded genes were significantly enriched in photosynthesis-related functions (DAVID Functional Annotation Clustering; $p < 0.01$) (Supplementary Figure 5). Most divergent responses are either positive or negative across all treatments; however, we identified a total of 56 proteins where at least one treatment exhibited a response in a different direction from the other treatments (Supplementary Figure 5). Enriched biological functions within this group include photosynthesis (five proteins) and ribosomal proteins (six proteins) (DAVID Functional Annotation Clustering; $p < 0.01$).

**Metabolic context of the protein expression profiles.** Overlaying the protein expression data on KEGG metabolic diagrams shows that the strongest proteome-level changes are clustered around the metabolic pathways of photosynthesis, glycolysis/gluconeogenesis, and TCA cycle (Fig. 4). Specifically, the Calvin–Benson cycle and

gluconeogenetic pathway are near-consistently upregulated across all treatments, while glycolysis and TCA cycle are downregulated. Strong upregulation is also prominent among ribosomal components and photosynthetic machinery (Supplementary Data 1). Proteins related to glycolysis (Fig. 4) and TCA cycle (Fig. 4; shaded in pink) are generally downregulated under non-substitutable resource limitation and salt stress, suggesting that catabolism of carbohydrates is suppressed relative to the ancestors and controls. On the other hand, Calvin–Benson cycle (Fig. 4; shaded in gray) is active and producing glycerate-3-phosphate (G3P), which is channeled up the gluconeogenetic pathway into glyceraldehyde-3-phosphate (Fig. 4; shaded in brown). Glyceraldehyde-3-phosphate is partly being used to maintain an active Calvin–Benson cycle but is also channeled toward production of α-D-glucose-6-phosphate, which is a starting point of further anabolic reactions such as starch synthesis. The details of further anabolic activity are less clearly supported by our data; for instance the data show no upregulation of the synthesis of amino acids, fatty acids, and nucleotides (Supplementary Data 1), and it remains unclear whether the cells are synthesizing starch (Fig. 4; shaded in cyan). We observe a divergent response in the low-phosphate treatment, which shows an increased conversion of L-methionine to S-adenosyl-L-methionine and the conversion of S-adenosyl-L-homocysteine to L-homocysteine (Supplementary Data 1). However, this response has no obvious relation to increased phosphorus uptake or metabolism.

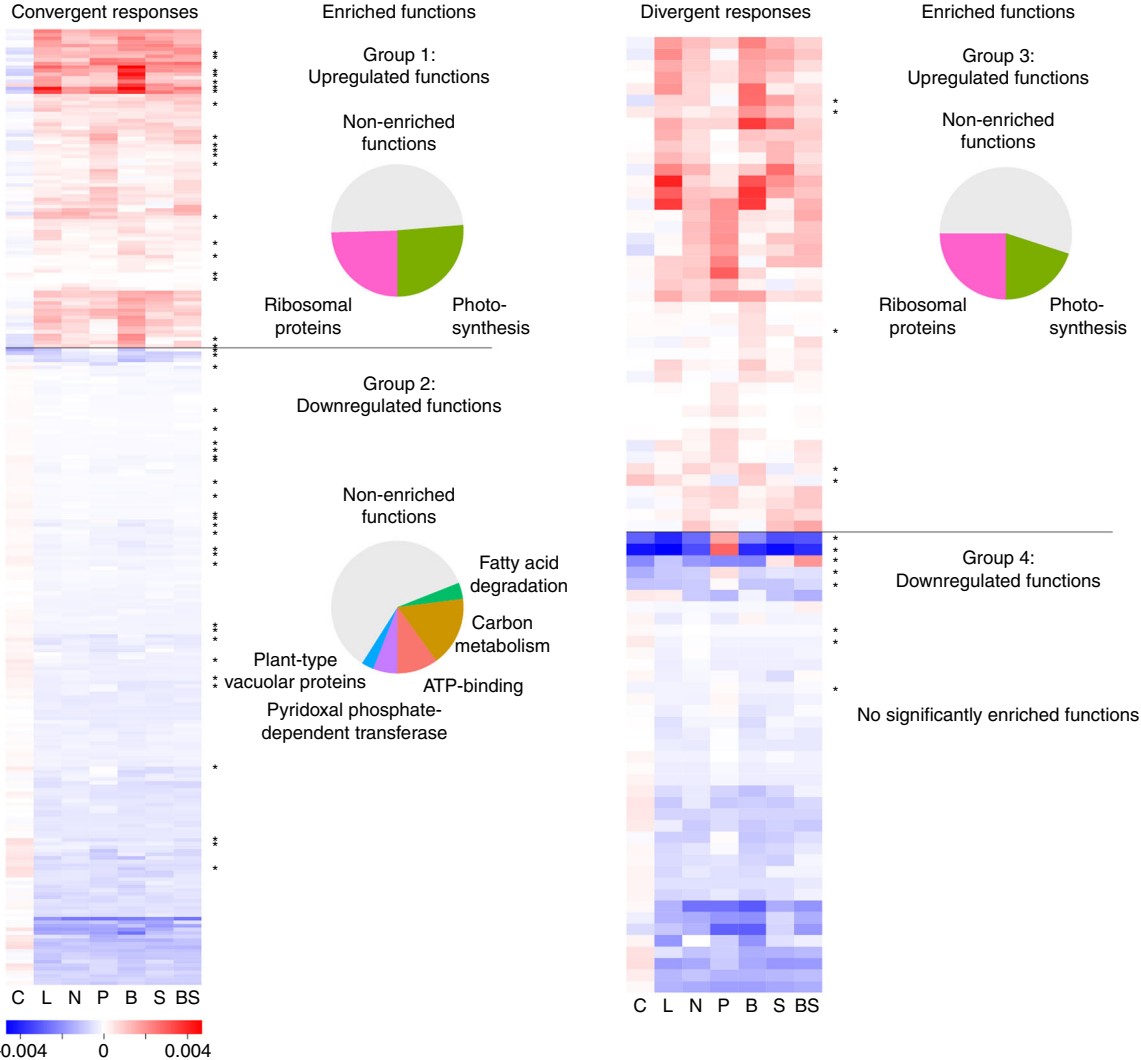

**Fig. 3** Protein expression level heatmaps and annotation for the differentially expressed proteins. Both divergent and convergent responses were divided into upregulated and downregulated functions based on the mean expression level of each protein across the treatments (marked below the heat maps). Annotation from GO Biological Processes was available for 166 of 351 proteins. Positive convergent and divergent responses were significantly enriched in photosynthesis- and ribosome-related proteins (DAVID Functional Enrichment Analysis $p < 0.01$). Negative convergent responses were significantly enriched in proteins related to carbon metabolism, ATP binding, pyridoxal phosphate-dependent transferases, fatty acid degradation-related proteins, and plant-type vacuolar proteins (DAVID Functional Enrichment Analysis $p < 0.01$). Negative divergent responses had no significantly enriched proteins. Additionally, we note that two proteins related to methionine metabolism are upregulated in the low-phosphate (P) treatment but downregulated in all others. An asterisk (*) indicates that the control is significantly different from zero ($t$ test $p < 0.05$), indicative of adaptations to chemostatic conditions

To determine whether these evolutionary changes in protein expression influence population-level phenotypes, we grew the ancestors and descendant populations up in the same common garden environment used for the proteomics experiment and measured a number of variables related to carbon and nutrient metabolism: photosynthesis, respiration, the stoichiometry of biomass carbon, nitrogen and phosphorus, as well as total biomass production and nutrient uptake per unit biomass. We found that the descendants of the evolution experiment under resource-limitation and salt stress selection treatments tended to have greater biomass carbon to phosphorus and/or carbon to nitrogen molar ratios (Supplementary Figure 6). This finding is consistent with the observed decreased phosphate uptake per unit of biomass produced in the low-phosphate and biotic treatments (although we found no-effect on the uptake of nitrogen) (Supplementary Figure 7). Respiration rates decreased across the selection treatments, in line with the observed decrease in the expression of many mitochondrial proteins. However, we also observed a decrease in photosynthetic carbon assimilation in the low-nitrogen and biotic treatments (Supplementary Figure 8).

## Discussion

Convergent evolution is typically observed in cases where organisms evolve in similar conditions[11,32] or where divergent selection is weak relative to dispersal, permitting the evolution of a single-generalist phenotype[20]. On the other hand, it has been reported that different nutrient limitation scenarios lead to a variable range of evolutionary outcomes, with some resource-limitation environments enabling a broad range of outcomes, and others resulting in few, apparently constrained, adaptive solutions[11]. Here, we report convergent proteomic evolution across variable selective environments. In our experiment, each of the resource-limitation treatments (low-light, low-nitrogen, and low-phosphorus) was fundamentally different from any other, because we imposed limitation of different, non-substitutable resources.

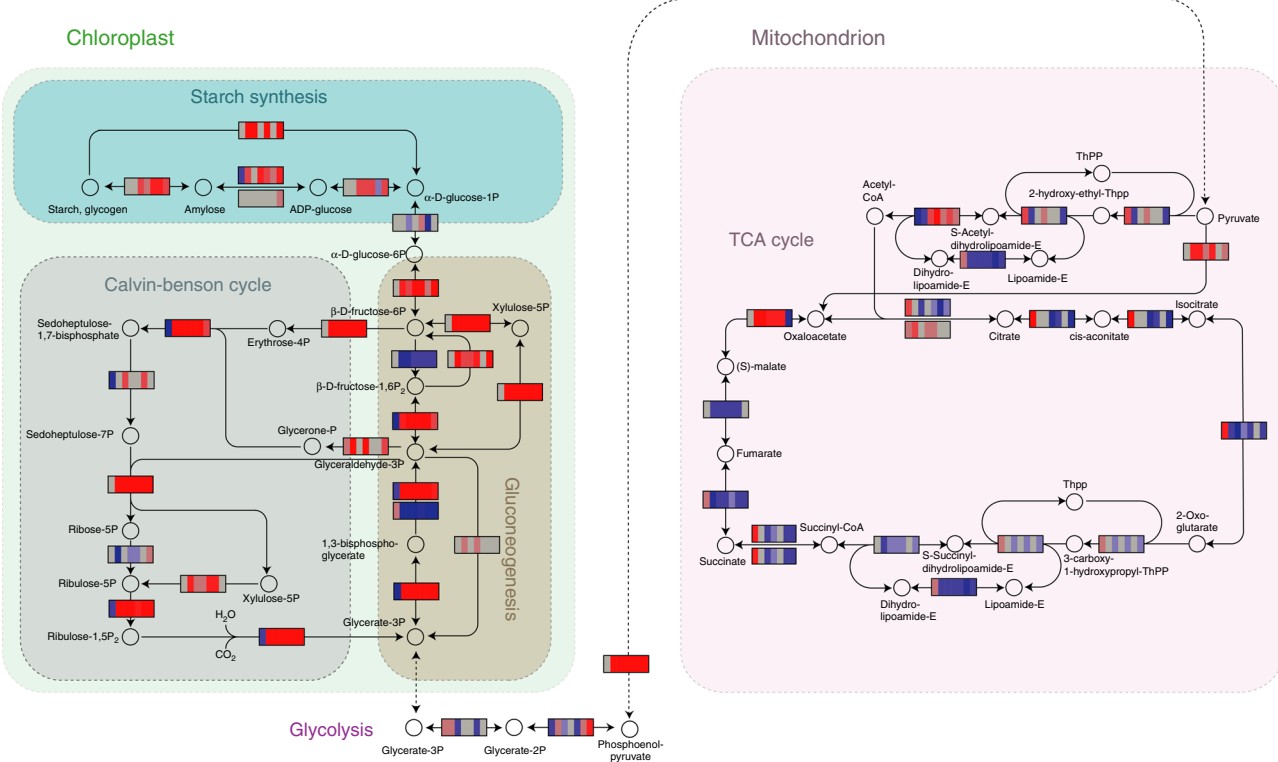

**Fig. 4** Core metabolic pathways of *C. reinhardtii*. The circles represent metabolites, and the rectangular heat maps the enzymes, with blue indicating downregulation, red upregulation, and gray no significant change. The order of the colored boxes within the heat maps correspond to the selection treatments (as in Fig. 3), and the protein expression scale corresponds to the normalized peptide intensities. Metabolic functions are indicated by shading in the following colors: Calvin–Benson cycle (gray); Starch synthesis (cyan); Gluconeogenesis (brown); TCA cycle (pink). The most consistent response among the different selection treatments is the upregulation of anabolic functions including of the photosynthetic machinery, Calvin–Benson cycle and gluconeogenetic pathways. The Calvin–Benson cycle produces glycerate-3-phosphate (G3P), which is likely channeled up the gluconeogenetic pathway into α-D-glucose-6-phosphate. By contrast, catabolic functions of the TCA cycle (for respiration) and the conversion of G3P into glycerate-2-phosphate are being mostly downregulated

Furthermore, each selection treatment was kept discrete from all other treatments, allowing no dispersal between the treatments which precludes selection for a generalist, adapted to all resource-limitation environments. Finally, the salt-stress treatment provided a point of comparison with investigate unique adaptations to resource-limitation as opposed to stress in general. As a result, it does not seem likely that the convergent evolution we observed across different types of resource-limitation and salt stress, could have resulted from either evolution to similar experimental conditions, or from the evolution of a generalist phenotype via dispersal.

The upregulated functions across the treatments include very central, highly conserved metabolic functions, such as photosynthesis and protein synthesis[33]. Convergent selection of such central functions supports our hypothesis that non-substitutable resource limitation of any kind obstructs core metabolism, growth and the completion of the cell cycle, and therefore directs evolutionary responses to optimize the upregulation and downregulation of proteins related to core metabolic functions. We interpret this optimization as convergent evolution across resource-limitation environments because regardless of resource identity, improvements in metabolic efficiency can only result from a limited set of core metabolic pathways. The lack of distinct proteomic responses among the salt-stress treatments supports our hypothesis that these adaptations are convergent responses to alleviate obstructions of core metabolic pathways.

This convergent evolution proteomic response was also associated with an improvement in the resource assimilation efficiency of nitrogen and phosphorus into biomass. We observe no

evidence for increased acquisition of any limiting resources, observed for instance in refs. [16–19]. On the other hand, upregulation of the core metabolic pathways of Calvin–Benson cycle and gluconeogenesis, and other core functions, such as photosynthesis and protein synthesis[33], strongly support our hypothesis that the cells use increased metabolic efficiency as an adaptive strategy to non-substitutable resource limitation. This hypothesis is further supported by our observation that the descendants of the evolution experiment under resource-limitation and salt-stress selection treatments tended to have greater biomass carbon to phosphorus and/or carbon to nitrogen molar ratios, when grown in the same common garden environment as used for the proteomics experiment. Since the majority of phosphorus in cells is contained in the rRNA molecules of the ribosomes, and most of the nitrogen is contained in the protein-rich light-harvesting machinery of the chloroplasts[34,35], the elevated biomass molar C:P and C:N ratios indicate that more carbon is fixed per ribosome or per chloroplast within the cells, respectively[36–38]. The similarity of the responses between the salt stress and resource-limitation environments suggest that protein expression evolved so as to mitigate the obstruction of core metabolic pathways.

Despite a general increase in the net assimilation of carbon relative to other limiting resources (e.g., nitrogen and phosphorus), we did not observe an overall increase in the rate of net carbon assimilation across the treatments. While respiration rates decreased across selection treatments, in line with the observed decrease in the expression of many mitochondrial proteins, we also observed a decrease in photosynthetic carbon assimilation in the low-nitrogen and biotic treatments. The absence of an

**Table 1 The temporal sequence of resource-limitation and salt stress experimental treatments**

| Month | Control (C) | Nitogen (N) (NaNO$_3$, μmol/L) | Phosphorus (P) (K$_2$HPO$_4$ μmol/L) | Biotic (B) (proportion biotically-depleted) | Light (L) (μM photons·m$^2$·s$^{-1}$) | Salt (S) (NaCl g/L) | Biotic & salt (BS) | |
|---|---|---|---|---|---|---|---|---|
| 1 | . | 1000 | 50 | 0 | 100 | 0 | 0 | 0 |
| 2 | . | 100 | 5 | 0.01 | 70 | 1 | 0.01 | 1 |
| 3 | . | 100 | 5 | 0.1 | 50 | 2 | 0.1 | 2 |
| 4 | . | 10 | 0.5 | 0.4 | 20 | 4 | 0.4 | 4 |
| 5 | . | 10 | 0.5 | 0.75 | 15 | 6 | 0.75 | 6 |
| 6 | . | 10 | 0.5 | 0.95 | 5 | 8 | 0.95 | 8 |
| 7 | . | 10 | 0.5 | 1 | 5 | 8 | 1 | 8 |
| 8 | . | 10 | 0.5 | 1 | 5 | 8 | 1 | 8 |

increase in rates of photosynthetic carbon fixation despite the upregulation of many chloroplast proteins, may be an example of evolutionary phenotypic buffering, or compensation, where the evolved changes in the molecular phenotype buffer the impacts of stress and serve to maintain levels of functioning at higher levels of biological organization (i.e., cellular photosynthesis)[39]. Overall, the ratio of carbon assimilation to respiration increased significantly only in the low-phosphate treatment, indicating a gain in net carbon metabolic efficiency only in this treatment. So, while carbon metabolism itself did not generally become more efficient under long-term low-resource selection, C:N and C:P biomass ratios tended to increase, indicating that nitrogen and phosphorus assimilation efficiency improved relative to net carbon fixation.

In conclusion, the signature of the evolutionary response of *Chlamydomonas reinhardtii* protein expression to non-substitutable resource limitation is one of improved metabolic efficiency. Rather than evolving an improved ability to acquire resources, the descendants of the evolution experiment displayed signatures of more efficient allocation and assimilation of resources, with an improvement in the resource assimilation efficiency of nitrogen and phosphorus into biomass. We found that adaptation to resource limitation involves significant changes in protein expression, which show a convergence across the resource limitation environments. Surprisingly, similar patterns of adaptation were also observed under salt stress. We propose that this is because both resource limitation and salt stress obstruct core metabolic pathways which essentially impose limits on cell growth and division, and therefore select for similar evolutionary outcomes. These findings represent a contrast to previous work which had investigated only the plastic responses of the proteome observed under resource limitation or salt stress (but see[11]). Interestingly, previous studies on plastic responses have generally reported the opposite patterns on modulation of core metabolic pathways to that reported here, supporting the hypothesis that plastic variation in the expression of phenotypes often occurs in the opposite to the direction of adaptive phenotype evolution[40]. The general agreement between conclusions drawn from the proteomic data and stoichiometry suggests that investigations of the evolution of the molecular phenotype may one day help to link our understanding of evolution across biological scales: from the genome to the molecular phenotype and beyond, to macroscopic phenotypes such as stoichiometry, and perhaps eventually to population- and community-level dynamics.

## Methods

**Evolution experiment**. The evolution experiment began from an inoculum of *Chlamydomonas reinhardtii* CC1690 wild type mt + obtained from the Chlamydomonas Center (http://www.chlamycollection.org/product/cc-1690-wild-type-mt-sager-21-gr/) in October 2014. Prior to the initiation of the evolution experiment,

the inoculum was maintained in the lab under sterile conditions, in semi-continuous batch culture in a liquid freshwater medium called COMBO[41]. Throughout the following experiment, we used a modified COMBO medium which did not contain silicon, vitamins, or animal trace elements because they were unnecessary for *C. reinhardtii*. We refer to this modified medium as "COMBO" throughout. We plated a dilute liquid batch culture onto agar after four months, and on February 19, 2015, we haphazardly isolated four clonal colonies (grown from single cells) from the agar plates and transferred them into COMBO to produce isoclonal liquid batch cultures. We named these clonal populations Ancestors 2, 3, 4, and 5 ("Anc2" etc. hereafter). Nine days later, on February 28, 2015, we inoculated the four clonal ancestors and the original population, CC1690, into seven chemostats each. The seven chemostats into which each ancestral population was inoculated were randomly assigned to a resource-depletion (or salt-stress) treatment (Table 1). In total, we inoculated 35 chemostats: 5 ancestral populations × 7 experimental evolution treatments. The experimental setup is described in Supplementary Figure 1.

All 35 chemostats originally contained 23 mL of sterile COMBO and were inoculated with 5 mLs of liquid batch culture. Chemostats were composed of autoclave-sterilized screw-top vials (Supelco™ 40 mL vials with hole cap and barrier/septa), each containing a magnetic stir bar, and three syringes which pierced through the Teflon-lined rubber lid. One syringe was under suction to produce negative pressure within the chemostat. The negative pressure allowed liquid sampling and air intake through another syringe. The second syringe, attached to an air filter, acted as an air inlet. The third syringe, attached to a media bottle, acted as a media inlet. All materials and media were autoclaved for sterility. All chemostats were maintained under ~90 μM photons·m$^2$·s$^{-1}$ of light on an 18 h light, 6 h dark light cycle at 20 °C ("standard conditions" hereafter). Chemostats were constantly stirred using magnetic stir bars and stir plates. Two peristaltic pumps were used: one to generate suction and continuously pull sterile air through 0.45 μm filters and into the medium, and the second to pump sterile medium into the chemostats.

After two weeks of growth in chemostat without media replacement, the chemostats began receiving daily exchanges of replete sterile COMBO via 20-min media exchanges conducted via peristaltic pump at 400 rpm. Daily dilution rates were 15.74 mL +/- 0.25 mLs, or 56% per day. On April 8, 2015, after 1 month of growth in replete COMBO, we initiated the resource limitation and salt stress regimes (Table 1). Dissolved inorganic nutrients were manipulated by reducing the availability of nitrate in the form of NaNO$_3$ or phosphate in the form of K$_2$HPO$_4$ in the medium being exchanged every day. We followed the standard protocol for low phosphate potassium ion replacement[41]. The light limitation regime was imposed by completely covering the surface of the chemostats with neutral-density light filter paper (Solar Graphics™, Clearwater, Florida) so as to filter out a fixed percentage of light across all wavelengths of the light spectrum. The salt stress was imposed by increasing the sodium chloride concentration in the COMBO. For reference, pilot studies showed that the ancestors had a salt tolerance at ~4 g L$^{-1}$. Each resource limitation or salt stress level was maintained for one month before the next level was imposed (Table 1).

The biotically-depleted medium (hereafter "Biotic") was produced by individually growing seven other species (Supplementary Table 1) of freshwater algae in batch culture in replete COMBO, and removing the phytoplankton biomass from each culture, sterilizing the medium, and mixing resultant spent media from each of the seven cultures. To do this, we first scaled batch cultures of each species up from 100 mLs to 4 L over the course of 6 weeks under standard conditions. The phytoplankton were then removed from the depleted medium by first centrifuging each culture at 5000 rpm for 15 m, keeping the supernatant, and then filtering the supernatant through 0.45-μm cartridge filters (Sartorius Stedim™ Sartobran® P 0.45 μm) using vacuum filtration. Depleted and filtered supernatants from all cultures were combined into a single 50-L carboy and autoclave-sterilized. After autoclaving, this spent medium was cooled and stored at 4 °C in the dark until further use.

At the termination of the evolution experiment, the chemostats were sampled and 5 mLs from every chemostat were inoculated into 10 mLs of replete COMBO

in batch culture. The cultures were then grown under standard conditions until visibly green (up to 3 weeks) before plating onto agar in triplicate. This culture step was performed in order to ensure that each descendant population had a large enough population density for successful plating, and that even very low-density selection lines had enough cells to ensure successful live sample storage. The agar-plated algae were again grown under standard conditions until green (for up to 3 weeks) before being transferred to a climate chamber maintained at 12 °C and ~ 40 μM photons·m$^2$·s$^{-1}$ for cold storage.

**Common garden growth and MudPIT**. Each algal population was removed from cold storage using sterile technique and was again inoculated into 50 mLs of replete COMBO under standard conditions in order to produce enough biomass for protein extraction and quantitation using LC-MS/MS (see below). In culturing the algae under standard conditions both before and after storage, we ensured that all differences in protein expression between ancestors and descendants were due solely to heritable genetic or epigenetic change, rather than due to maternal effects or plasticity induced by the historical selection environment. Each culture grew for 7 days before we centrifuged them at 4000 rpm, decanted the supernatant, and froze the pellets at −80°C.

Sample preparation for protein analysis was done following established procedures[42,43]. Briefly, frozen algae were thawed at room temperature before lysis, homogenized using a grinder, sonicated on ice (3 × 10 s with 30 s pause in-between) and finally centrifuged (13,000 rpm, 60 min, 4 °C). The supernatant was removed and centrifuged again using the same conditions (two repetitions). The proteins were then precipitated using methanol/chloroform and the formed pellet air-dried (20 min) and then re-dissolved in buffer, after improving solubility by wetting with 0.2 M NaOH. The protein amount in the solution was determined using the Bradford assay.

Prior to the tryptic digestion, 100 μg of protein were reduced using TCEP, and carboxy-amidomethylated using IAA. The treated sample was then put on a shaker and digested with trypsin overnight (ratio of 100:1, 37 °C, 14–16 h). The digestion was stopped by the addition of formic acid. After filtration (0.45 μm, Durapore PVDF, Merck Millipore), the samples were transferred to glass vials for storage or directly loaded onto a commercially available trap column (5 mm, 300 μm ID, 5 μm, 100 Å, C18 Acclaim PepMap 100, Dionex), using a nanoHPLC (Ultimate 3000, Dionex). Peptides were then eluted onto a SCX column (3.5 cm, 100 μm ID, 363 μm OD, BGB Analytik AG, in-house pressure filled with 5 μm, Nucleosil 100-5 SA from Macherey Nagel AG, and closed with a frit) and subsequently analyzed on a C18 column (4.5 cm, 100 μm ID, in-house pressure filled with 3 μm, 100 Å, Nucleodur C18 Pyramid, Macherey Nagel AG) pulled to a needle for electrospraying (Sutter Instrument, P-2000, Science Products AG Basel). The SCX and C18 columns were linked in series by a 25 μm ID fused silica capillary. Both the SCX and C18 columns were used for three analytical runs only and then replaced.

Peptides eluting off the C18 column were directly sprayed into the high-resolution mass spectrometer (LTQ-Orbitrap XL, Thermo Scientific, Bremen, Germany). The instrument was tuned and calibrated using angiotensin (Sigma-Aldrich) and a calibration mixture from Thermo Scientific (ProteoMass LTQ/FT-Hybrid ESI Pos. Mode Cal Mix, Supelco), respectively. The instrument was operated in positive ion mode (needle voltage: + 1.2 kV, tube lens: 135 V, ion transfer capillary: 200 °C). Using a standard 11-step MudPIT protocol, peptides were sequentially eluted onto the C18 column by eleven salt pulses (4.9 min, from 0 to 100% eluent C, in 10% increments, eluent C: 0.5 M aqueous NH$_4$Ac, 5% ACN, 0.1% formic acid), and then separated by gradient elution[42,43].

The LTQ-Orbitrap XL acquired scan-dependent MS/MS of the peptides eluting off the column. For this the instrument selected the 7 most intense ions found in a mass spectrum acquired in a pre-scan (300–1600 m/z, resolution of 7500), then analyzed them in the linear ion trap using MS/MS with a normalized collision energy of 35%, while in parallel acquiring a high-resolution full scan in the orbitrap (300–1600 m/z, resolution of 60,000). When selected ions had been analyzed twice, with a minimal signal intensity of 1000 counts they were excluded from reanalysis for 60 s, a procedure called dynamic exclusion. The HPLC and mass spectrometer were controlled by Xcalibur (Thermo Scientific). Each ancestral sample was run in three technical replicates, while descendant samples were only run once.

**Peptide count estimation**. Peptide count estimates were extracted from the raw data using the MaxQuant (version 1.5.7.4) software package[44], with match between runs enabled and otherwise using the standard developer recommended parameters for label-free proteome quantification (config files mqpar_v16.xml and parameters.txt available at https://github.com/manutamminen/chlamy_proteome_evolution).

The peptide searches were performed using Andromeda[45] using *Chlamydomonas reinhardtii* reference genome assembly version 5.0 and annotation version 5.5 from Phytozome[46] and NCBI Genbank entries U03843.1 and BK000554.2 for mitochondrial and chloroplast references, respectively. The maximum permitted number of missed trypsin cleavages was set to 2. The analysis was set to consider carbamidomethyl of cysteine as a fixed modification and oxidation of methionine and acetylation of the protein N terminus as variable modifications. Mass tolerance for precursor and fragment ions were set to 20 ppm for the first search and 4.5 ppm for the main search. The following criteria were

applied for peptide identification: Andromeda *p*-value threshold < 0.05, minimal peptide length 7, minimal score of unmodified peptides 0, minimal score of modified peptides 40, peptide FDR (false discovery rate) cutoff 0.01, and protein FDR cutoff 0.01. The Andromeda *p*-value threshold was defined as the probability *P* of matching at least k out of n theoretical masses in the peptide search database is calculated. Thus, -10*log(*P*) gives the Andromeda "*p*-value". Peptide FDR was calculated as follows: first, forward, and reverse(^ = decoy) protein databases are constructed. The spectra are matched against both databases. The resulting Andromeda match score histograms were used to estimate the continuous score distributions via kernel smoothing. The score distribution estimates were then used to calculate the posterior error probability (PEP) for each peptide. Protein FDR was calculated as follows: first, for each peptide in a protein group, the spectrum with the lowest FDR was selected. Then, all selected peptide FDRs were multiplied to get the Protein FDR. For the abundance estimates, the sum of the unique and razor peptides were used, where the latter is calculated by assigning a group of peptides to the protein with the highest number of matching identified peptides.

**Biomass and stoichiometry**. All ancestral and descendant populations were sub-cultured into 50 mL of liquid COMBO from agar plates that had been maintained in cold storage, and underwent a growth period of 7 days before the start of the experiment. Due to significant fungal or bacterial contamination, three populations were lost from the 40 ancestral and descendant populations analyzed in the proteomics experiment: namely the controls of Anc2 and Anc3, as well as the biotic and high-salt treatment (BS) of Anc5. We then diluted all cultures to achieve the same, low starting density across all populations before inoculating them into the stoichiometry experiment. Our target was a diluted concentration of 1000 raw fluorescence units (RFU, or raw fluorescence of cells in the range of chlorophyll-a), measured on a plate reader, which we used as a proxy for cell density. We carried out dilutions using autoclaved COMBO in a laminar flow hood to ensure sterility. On the next day, we inoculated 1 mL of each diluted culture into 170 mL of COMBO medium, and grew them under standard conditions for 7 days. During this time, we sampled 1 mL from all cultures daily and measured the RFU on a plate reader in order to monitor the growth trajectories. On the 8th day, when all growth curves were approaching saturation, we harvested the algal biomass by filtering each culture onto a set of two ashed (400 °C) and pre-massed Whatman ® glass microfiber filters (grade GF/F 47 mm). The filters were then dried in an oven overnight at 60 °C, and post-massed to obtain an estimate of total dry biomass per mL of culture filtered. One filter from each population was used to estimate the elemental carbon and nitrogen content of the biomass on an Elementar vario PYRO cube EA-IRMS, and the other filter was used to estimate phosphorus content using Skalar San + + Continuous Flow P/N analyser. The phosphorus samples were first digested and completely oxidized using a peroxydisulfate solution. Digested samples were diluted 1:20 before being run on the P/N analyser.

**Chlorophyll, photosynthesis, and respiration measurements**. As for the biomass, stoichiometry and nutrient experiments, populations were sub-cultured into liquid COMBO from agar plates that had been maintained in cold storage. The cultures then underwent a growth period of 7 days before the start of the stoichiometry experiment. Three populations were lost from experiment, as for the biomass, stoichiometry and nutrients experiment: the controls of Anc2 and Anc3, as well as the biotic and high-salt treatment (BS) of Anc5. After the acclimation period, we performed dilutions in order to achieve the same low starting density across all populations, with a target diluted concentration of 1000 raw fluorescence units, measured on a plate reader. All dilutions were carried out using autoclaved COMBO in a laminar flow hood to ensure sterility. On the day after dilutions, we inoculated 1 mL of each diluted culture into 170 mL of COMBO, and grew them under standard conditions for 9 days. On the 10th day we collected a 40 mL and a 50 mL sub-sample of each culture for chlorophyll and respiration measurements, respectively. We also took 50 mL samples for the estimation of photosynthetic carbon assimilation, which we diluted one in two before analysis to ensure that the method would be within the upper detection limit of the method.

We filtered 40 mL of each culture onto 47 mm GF/F filters (Whatman) for chlorophyll analysis. We then folded the filters into 15-mL conical tubes and froze them at −20 °C. Four days later they were extracted in 8 mL of 90% ethanol. We then vortexed and sonicated the samples in a water bath at 20 °C for 15 min. We incubated the extracted samples overnight in the dark at 4 °C. We then filtered the samples through 0.2-μm cellulose acetate filters (Sebio) to remove any glass filter particles. The concentrations of chlorophyll-a, -b, and lutein were measured in an eluent of 49.5% methanol, 45% ethyl acetate, and 5.5% water on a LichroSpher 100 RP-18 HPLC column (Merck & Co. Inc., White House Station, NJ, USA) at a flux rate of 1.0 ml min$^{-1}$. The column was connected to a Jasco AS2055 Plus auto-sampler and PU 2089 plus liquid chromatography pump. Peaks were identified using retention times and spectrum analysis. Standards for chlorophyll-a, -b, and lutein (among others) were purchased from DHI, Denmark.

We measured respiration by estimating the rate of oxygen depletion in each sample over a dark incubation period of one hour. We performed oxygen measurements in four blocks of up to 10 samples, randomized over time. Samples were incubated in custom-made "respirometers", each one consisting of a Schmizo glass vial fitted with a motorized mixing attachment, a PreSens Oxygen-sensitive

spot, and a fiber-optic cable to transfer data to a PreSens Fi box. We carefully lowered the mixing attachment and sensing cable into each sample (50 mL) to ensure the absence of air bubbles while sealing the sample via an O-ring. Samples were maintained at 20 °C via a cooling block. Before the measurements began, all oxygen-sensitive spots were calibrated to 0% oxygen at 20 °C. This measurement took place in sodium bisulfite-treated water. Subsequently, all sensors were calibrated in oxygen-saturated water, also at 20 °C. We estimated respiration as the slope of the measured oxygen over 1 h, removing the first 20 min of measurements in which the oxygen measurements settled on a stable slope.

To estimate photosynthetic carbon assimilation, we first measured alkalinity using titration with HCl[47]. We then determined the total inorganic carbon (TIC) from the alkalinity and pH measurements[48]. We then measured primary production by determining the rate of in situ $^{14}C$ assimilation, using the acid bubbling method according to ref. [48]. For each phytoplankton population, we spiked a 40 mL sample with 5µCi NaH$^{14}$CO$_3$ in a 50 mL conical tube. From these 40 mL, 7 mL was subsampled and placed into transparent glass Scintivials and incubated in the light at 90 µmol·m$^{-2}$·s$^{-1}$ and 20 °C for 3.5 h. A second, 7 mL sub-sample was transferred into Scintivials and incubated in the dark at 20 °C for 3.5 h. Finally, the last 7 mL sub-samples were transferred into Scintivials with 10 mL of Instagel Plus$^{TM}$ (Packard, USA). After incubation, the light- and dark-incubated vials were acidified with 100 µL of 6 N HNO$_3$ for ca. 50 min, and then 10 mL of Instagel Plus$^{TM}$ (Packard, USA) was added. We then determined the radioactivity of the sub-samples in a liquid scintillation counter (Model Tricarb, Packard, USA). We calculated carbon assimilation per chlorophyll (mg C· mg chlorophyll$^{-1}$·h$^{-1}$). The fraction of $^{14}C$ that was assimilated multiplied by the TIC in the sample corresponds to the instantaneous carbon assimilation ($P_B$). Furthermore, all $P_B$ values were corrected for non-photosynthetic fixation of carbon by subtracting $P_B$ determined in the dark sub-sample. Details of the method are further described in refs. [49,50].

**Peptide count processing, statistics, and visualization**. The abundance effects within the peptide counts were accounted for by dividing each peptide intensity with a marginal sum across each treatment and population, resulting in normalized peptide intensities, using pandas version 0.19 in Python 3.5. The effect of treatment and population on the peptide intensities were visualized and tested using RDA (with 51 independent observations and 2 and 3347 dimensions for the explanatory and response matrices, respectively), implemented in vegan version 2.4[51]. The group significances were tested using permutational anova using populations as replicates and conditioning by Treatments. The actual analysis was performed using capscale and anova.cca functions implemented in R-package vegan. Normality of the fitted values was tested by Shapiro tests, implemented in R-package stats as a function shapiro.test. Significance of the treatment differences across the two first RDA axes were tested using ANOVA and Tukey's honestly significant difference tests implemented in R-package stats as functions anova and TukeyHSD.

Significantly upregulated or downregulated proteins relative to the ancestors were identified by subtracting the ancestral means of the normalized peptide intensities from the corresponding population values, and testing these differences from zero using one-sample $t$ tests. Normality of the expression levels was tested by Shapiro tests, implemented in R-package stats as a function shapiro.test. Treatments that are significantly different from the control were identified using Dunnett's tests, implemented in R package multcomp version 1.4 as function glht. Convergent and divergent proteomic responses were identified using Friedman's tests, implemented in R package stats as a function friedman.test. These tests were performed on R version 3.3.2. The population-aggregated means of the proteins were visualized using Circos version 0.69[52]. Heatmaps were drawn using heatmap.2 function implemented in R package gplots version 3.0.1 in R. The associated hierarchical clustering was drawn using default distance and clustering options of heatmap.2. All other image preparation was performed using ggplot2 version 2.2.1[53] in R, and superficially edited on Adobe Illustrator version 21.1.0. The detected proteins were annotated using data from Gene Ontology Biological Processes, provided by PANTHER database[54]. Enrichment analyses were performed using the default options of the Functional Annotation Clustering Tool provided by DAVID Bioinformatic Resources version 6.8[55,56]. The metabolic pathways were overlaid on KEGG metabolic pathways using Pathview version 3.5 provided by R/Bioconductor[57]. Significance of each phenotypic measurement was determined by comparing each treatment to the corresponding Ancestor using two-sided Wilcoxon tests, implemented in R-package stats as a function wilcox.test. $P$-values were adjusted for false discovery rate (method "fdr") using function p. adjust implemented in R-package

**Code availability**. The source code of the analyses is available at https://github.com/manutamminen/chlamy_proteome_evolution.

## Data availability

Proteomic data that support the findings of this study have been deposited in PRIDE PRoteomics IDEntifications database[58] under the identifier PXD010847. The MaxQuant-processed data, the phenotypic measurements and other relevant data is provided together with the source code at https://github.com/manutamminen/chlamy_proteome_evolution.

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

## Acknowledgements
We thank S. Gut, P. Ganesanandamoorthy, N. Minas, G. Siegrist for their help in maintaining the chemostats and *C. reinhardtii* cultures. We thank R. Schönenberger for training and supervision in running the proteomics samples on the LC-MS/MS. We thank Daniel Steiner for performing the phenotypic measurements. We thank S. Dennis, P. Feulner, J. Jokela, and A. Zupanic for their insightful comments on the paper. This work was supported by an Eawag Postdoctoral Fellowship and an Eawag Seed Grant to AN.

## Author contributions
M. Tamminen analyzed the data, prepared the figures, and wrote the manuscript. A.B. processed the raw proteomic data into peptide counts. M. Thali processed all proteomic samples and edited the manuscript. A.P. prepared cultures for stoichiometric, photo-synthesis and respiration estimates, and edited the manuscript. B.M. helped to conceive of the experiment, develop strategies for data analysis, and edit the manuscript. M.S. developed the proteomics pipeline and edited the manuscript. A.N. conceived of and ran the evolution experiment, guided data collection and analysis, and wrote the manuscript.

## Additional information

**Competing interests:** The authors declare no competing interests.

