## [Peer Review File · Nature Communications]

Reviewers' comments:

Reviewer #1 (Remarks to the Author):

The authors measure protein abundance in strains of *Chlamydomonas reinhardtii* that have been evolved in different environments. They specifically contrast evolution in replete media against evolution in media with low nutrients or high salt. The authors find that cells evolved in the stressful environments tend to show similar changes in protein abundance relative to the control (replete media). The authors assert that the patterns are consistent with evolution increasing efficiency rather than uptake under nutrient limitation, and highlight that this is an interesting case of convergent evolution in different environments. This work addresses an interesting question of the extent to which evolution is repeatable and predictable across environments.

This paper would be greatly improved by adding data on growth phenotypes. The authors argue that the observed changes in protein abundance are consistent with optimizing efficiency of nutrient use over uptake rate but then never directly measure either of these properties. It would also be interesting to know whether adaptation to one stress actually improves fitness in the face of another. If there has been convergent adaptation to stress one would expect to see evolution increasing fitness across environments.

Related to this point, it is unclear to me why the authors focus solely on adaptation to nutrient limitation. The fact that salt treatments give similar patterns to nutrient limitation seems to suggest that changes are less related to nutrients and more related to stress in general. Additionally the "biotic" media is difficult to interpret. While other algae will draw down some nutrients, they are certainly excreting many others into the media. It is difficult to know what nutrients might be limited. Further, some of the excretions could be toxic so perhaps stunted growth in spent media is the result of toxins rather than nutrient limitation. If the desired message is about nutrient limitation, it seems that focusing on the 3 cases of known nutrient limitation would be stronger.

Minor comments:

- 1) It would be useful to know roughly how many generations occurred during the experimental evolution.
- 2) Gresham et al (PMC2586090) previously showed that different nutrient limitations lead to evolution of different gene expression in yeast. It would be useful to compare and contrast current results with this previous work.
- 3) I find figure 2A hard to read. I appreciate the attempt to provide readers with an overview of all data, but this advantage is lost if the data becomes too small to read.

Reviewer #2 (Remarks to the Author):

In the present manuscript the authors employed experimental evolution of the green alga *Chlamydomonas reinhardtii* under multiple different non- substitutable resource limitation regimes in combination with quantitative proteomics to investigate evolutionary adaptation of molecular phenotypes. Their data suggest that *C. reinhardtii* evolved to non-substitutable resource limitation via altered metabolic efficiency. In accordance, adaptation to resource limitation involved significant changes in protein expression showing similar patterns despite different non- substitutable resource limitation regimes. This is an interesting observation, which should be on the other hand accompanied with metabolic and physiological measurements. For the moment it is unclear, whether those significant expression changes observed, account for similar phenotypic outcomes such as photosynthetic capacity and CO₂ uptake, metabolic fluxes and growth performances and whether those changes result indeed in improved metabolic efficiency, as claimed in the conclusion. In accordance, additional experimental data should be provided to substantiate their conclusions and demonstrate whether and how metabolic efficiency has evolved. Having this wonderful resource in hands, it should be straightforward to provide these additional data.

Other comments.

1. In regard to LFQ, the authors write about spectral counts, do they mean peptide intensities? Did the author's also use spike-in-peptide controls for normalization?
2. Did the authors search for "mutated" peptides? "Mutated" peptides would not be identified and could influence due to absence protein abundances.
3. Fig. 4, please explain the protein expression scale in the figure legend.
4. Fig. 4, which protein expression is significant different in more than one treatment?
5. What about post-translational modifications, where PTM's found to be different in the different treatments?

Reviewers' Comments to Authors:

Reviewer 1

R1: This paper would be greatly improved by adding data on growth phenotypes. The authors argue that the observed changes in protein abundance are consistent with optimizing efficiency of nutrient use over uptake rate but then never directly measure either of these properties.

Our reply: This question from R1 is echoed by a similar question by R2. Essentially both reviewers requested some measurement of macroscopic phenotypes which would be able to support or reject our conclusion of increased metabolic efficiency rather than improved resource uptake. To address this request we ran a new set of experiments using the ancestors and descendants of the evolution experiment. We took all of the populations out of cold storage, putting them back into liquid culture. We then grew each population from very low density to saturated population growth, over the course of 8 days, following exactly the same methods used to generate the biomass for the proteomics experiments. We then harvested the biomass on the 8th day of the experiment and estimated biomass molar carbon, nitrogen and phosphorus, using an elemental analyzer and IRMS.

We estimated the stoichiometry of biomass for the ancestors and descendants, because it is an estimate of the amount of carbon fixed in the biomass relative to the units of biosynthetic machinery (e.g. ribosomes) and light-harvesting and photosynthetic machinery (e.g. chloroplasts) in the cells. The bulk of phosphorus in cells is contained in the rRNA molecules in the cells' ribosomes. These biomolecules are phosphorus rich, and are needed for protein synthesis and cell division (Sternier & Elser 2002; Loladze & Elser 2011). The bulk of the nitrogen in phytoplankton cells is contained in the protein-rich light-harvesting machinery, needed for photosynthesis (Sternier & Elser 2002; Loladze & Elser 2011). So broadly, C:P and C:N biomass ratios can be considered as estimates of the carbon fixed per ribosome or per chloroplast, respectively (Shuter 1979; Clark *et al.* 2013; Daines *et al.* 2014).

The biomass stoichiometry demonstrates that the non-control selection lines have all optimized their metabolism for efficient carbon assimilation relative to phosphorus, and also, in most treatments, relative to nitrogen. Had the populations improved resource uptake, we would not have expected greater biomass carbon to phosphorus or carbon to nitrogen ratios. We have now included this data as a new supplementary figure (Fig. S08). We wish to keep the focus in this paper on the evolved changes in proteome expression, rather than on other phenotypic changes, and so have included the figure, but only in the supplementary information. Nevertheless, we do feel that the data increases the strength of our conclusions, and for that reason we describe these findings in the Discussion on line 327-337. We are grateful for the constructive suggestions, which pushed us to go one step further in confirming our conclusions regarding the impacts of the proteomic changes on cellular metabolic efficiency.

R1: It would also be interesting to know whether adaptation to one stress actually improves fitness in the face of another. If there has been convergent adaptation to stress one would expect to see evolution increasing fitness across environments.

Our reply: This is an interesting and important question. In this paper however, our focus was on how protein expression evolved in the descendants relative to the ancestors. This scale of proteomic profiling from an evolution experiment has rarely, if ever, been accomplished. Running experiments to estimate fitness components in all ancestral and descendant environments is worthwhile, but we

consider this to be a separate set of experiments focused on a different set of scientific questions, which would distract from our analysis of proteomes.

R1: Related to this point, it is unclear to me why the authors focus solely on adaptation to nutrient limitation. The fact that salt treatments give similar patterns to nutrient limitation seems to suggest that changes are less related to nutrients and more related to stress in general.

Our reply: We have revised the manuscript to better illustrate our motivation for including the salt stress – specifically: 1) to provide an outlier source of selection with which to compare the effects of low resource selection and 2) to accelerate the adaptation via increased mutation rates induced by stress. Clarifying text has been added on lines 123-129, 221-223, 255-260, 296-298, 327-329 and 346-349. These include statements regarding how our conclusions regarding adaptation to non-substitutable resource limitation are influenced by a comparison to observed adaptation to another kind of stress, namely salt (lines 221-223, 255-257, 296-298).

R1: Additionally the "biotic" media is difficult to interpret. While other algae will draw down some nutrients, they are certainly excreting many others into the media. It is difficult to know what nutrients might be limited.

Our reply: We included the biotic medium to simulate the presence of a community of competitors which simultaneously deplete multiple nutrients. None of the species in this study are known to produce allelopathic or toxic substances, but nevertheless, it is possible that the growth of biotic competitors released organic compounds into the medium. Still, we estimated biomass densities over the course of the evolution experiment, and can confirm that the biomass of *Chlamydomonas* declined in response to increasing provision of “biotic” medium in the biotic treatments. This evidence supports the inference that that the evolving “biotic” populations became increasingly limited by the competitive effects of the other species, most likely due to resource depletion, over the course of the experiment (Response Figure 1).

Response Figure 1. Raw fluorescence units (y-axis) of five *C. reinhardtii* cultures over the course of the experiments (x-axis in days). A) Control treatment B) Biotic treatment.

R1: Further, some of the excretions could be toxic so perhaps stunted growth in spent media is the result of toxins rather than nutrient limitation.

Our reply: As noted above, to our knowledge, none of the species used for biotic depletion are known to produce allelochemicals or release toxins.

R1: If the desired message is about nutrient limitation, it seems that focusing on the 3 cases of known nutrient limitation would be stronger.

Our reply: Our question was not only about adaptation to nutrient limitation, but, slightly more generally about adaptation to limiting resources. In the evolution experiment, we simulated the effects of resource limitation, including for light (not a nutrient) by reducing resource availability. In the case of each individual resource (light, nitrogen and phosphorus) we imposed limitation by determining the exact level of each resources' availability. However, we also aimed to simulated resource limitation that may occur when species are competing with multiple competitors, which may each be better competitors for different resources. Though salt is not a resource for phytoplankton, we included salt stress in the evolution experiment for two reasons: 1) to provide an outlier source of selection with which to compare the effects of low resource selection, and 2) to accelerate the adaptation via increased mutation rates induced by additional stress. We have revised the manuscript to provide a clearer motivation both the salt stress and biotic treatments. Clarifying text has been added around lines 123-129, 221-223, 255-260, 296-298, 327-329 and 346-349.

Minor comments:

R1: 1) It would be useful to know roughly how many generations occurred during the experimental evolution.

Our reply: The experimental evolution took place over approximately 285 generations. This has now been indicated on line 129 in Materials and Methods as well as in Figure S01.

R1: 2) Gresham et al (PMC2586090) previously showed that different nutrient limitations lead to evolution of different gene expression in yeast. It would be useful to compare and contrast current results with this previous work.

Our reply: This is a great addition to the references, and relevant discussion has now been added Introduction (around lines 64-76), Results and Discussion (around lines 246-251) and Conclusions (around lines 350-351).

R1: 3) I find figure 2A hard to read. I appreciate the attempt to provide readers with an overview of all data, but this advantage is lost if the data becomes too small to read.

Our reply: We have simplified Fig. 2A by including only those proteins whose expression was significantly different from the Ancestors.

Reviewer 2:

R2: For the moment it is unclear, whether those significant expression changes observed, account for similar phenotypic outcomes such as photosynthetic capacity and CO2 uptake, metabolic fluxes and growth performances and whether those changes result indeed in improved metabolic efficiency, as claimed in the conclusion. In accordance, additional experimental data should be provided to substantiate their conclusions and demonstrate whether and how metabolic efficiency has evolved. Having this wonderful resource in hands, it should be straightforward to provide these additional data.

Our reply: This is a good suggestion, and in response we have run a set of experiments to estimate macroscopic phenotypes of the ancestors and descendants, measured at the same point in their growth curves, and under the same conditions as we used for the proteomics experiment. We

provide measurements of the stoichiometry of biomass, which we feel unambiguously demonstrate that the selection lines have optimized their metabolism for efficient carbon fixation relative other cellular components (i.e. phosphorus and nitrogen contained in ribosomes and chloroplasts, respectively). We provide this data as part of the paper, in a new supplementary figure, Fig. S08, and discuss the results on L. 329-337 of the Discussion. For further description, please see our response to the first comment from R1.

R2: 1. In regard to LFQ, the authors write about spectral counts, do they mean peptide intensities? Did the author's also use spike-in-peptide controls for normalization?

Our reply: The reported values, indicated in the previous manuscript version as spectral counts, can also be described as peptide intensities. This has now been corrected in the manuscript.

Here, we used MudPIT (Multidimensional protein identification technology) for the proteomic analysis, which does not involve peptide spike-ins.

R2: 2. Did the authors search for "mutated" peptides? "Mutated" peptides would not be identified and could influence due to absence protein abundances.

Our reply: Originally, we had not considered the influence of mutated peptides on the analysis. This is because an analysis of mutations is more complete when it is performed using genomic data, preferably whole genome re-sequencing, where mutations in non-coding and un-translated regions can be assessed. We do have an ongoing project on this, but the data analysis is arduous and is not yet ready for publication. Nevertheless, at the request of the reviewer, we pulled out the mutated peptides using MaxQuant, and found that the proportion of the mutated peptides in our dataset is low: 1,142 mutated peptides out of 1,417,605 non-mutated ones, or < 0.1%. We then re-ran the RDA analysis to include the mutated peptides, and found that our results are essentially unchanged (Response Figure 2, compare to manuscript Figure 1A). As a result, we provide this information for the review, but do not include it in the paper, where we aim to focus on changes in protein expression rather than in protein sequence.

Response Figure 2. RDA analysis performed using the expression levels of mutated and non-mutated proteins as response variables and Treatments conditioned by Strains as explanatory variables.

R2: 3. Fig. 4, please explain the protein expression scale in the figure legend.

Our reply: This has now been included in the legend of Fig. 4.

R2: 4. Fig. 4, which protein expression is significant different in more than one treatment?

Our reply: The significantly different treatments are now indicated in the figure.

R2: 5. What about post-translational modifications, where PTM's found to be different in the different treatments?

This is a very interesting and broad question, potentially deserving of a separate investigation. While such analysis is permitted by the MaxQuant analysis of the MudPIT data, we have chosen to omit it because in our opinion it would be a distraction from the current manuscript's focus.

1.

Clark, J.R., Lenton, T.M., Williams, H.T.P. & Daines, S.J. (2013). Environmental selection and resource allocation determine spatial patterns in picophytoplankton cell size. *Limnology and Oceanography*, 58, 1008-1022.

2.

Daines, S.J., Clark, J.R. & Lenton, T.M. (2014). Multiple environmental controls on phytoplankton growth strategies determine adaptive responses of the N : P ratio. *Ecology Letters*, 17, 414-425.

3.

Loladze, I. & Elser, J.J. (2011). The origins of the Redfield nitrogen-to-phosphorus ratio are in a homeostatic protein-to-rRNA ratio. *Ecology Letters*, 14, 244-250.

4.

Shuter, B. (1979). A model of physiological adaptation in unicellular algae. *Journal of Theoretical Biology*, 78, 519-552.

5.

Sterner, R.W. & Elser, J.J. (2002). *Ecological Stoichiometry. The biology of elements from molecules to the biosphere*. Princeton University Press, Princeton.

Reviewers' comments:

Reviewer #1 (Remarks to the Author):

I appreciate the extra experiments that the authors did to address my concerns. The changes in C:P and C:N ratios are interesting. It would be really nice if the authors could include some data about the amount of biomass produced by each strain (was biomass weighed during C:P measurements?) to support their assertion that biomass is being produced more efficiently. It is also unclear to me that it is possible to rule out changes in nutrient uptake given the data. While I buy that all treatments are altering expression of proteins associated with carbon metabolism, it is not clear that they are not also altering uptake kinetics. That being said it is interesting that parallel changes in protein expression are observed in the face of different stresses.

Figure 1 - On my screen (and printer) the control and salt treatment are both red. It would be useful to label each treatment a unique color.

Reviewer #2 (Remarks to the Author):

The manuscript has improved. However, some questions remain.

The new measurements provide information about the stoichiometry of biomass, indicating that the selection lines have optimized their metabolism for efficient carbon fixation relative other cellular components. Here, still the question remains how this could be explained, whether this is due to changes of photosynthetic capacity and CO₂ uptake or metabolic fluxes. Although her authors argue that the focus of the paper is on the proteomic analyses, these data are required for a journal with broader impact. The measurements of photosynthetic capacity and CO₂ uptake should be straightforward.

In regard to the mutated peptides, the authors used MaxQuant and found 1,142 mutated peptides out of 1,417,605 non-mutated one. Do these mutations cluster certain functional entities? Could the mutations explain phenotypic differences?

Reviewers' comments to the authors:

Reviewer 1:

R1: I appreciate the extra experiments that the authors did to address my concerns. The changes in C:P and C:N ratios are interesting. It would be really nice if the authors could include some data about the amount of biomass produced by each strain (was biomass weighed during C:P measurements?) to support their assertion that biomass is being produced more efficiently.

Yes, we had performed measurements of biomass at the same time as we had performed the previous stoichiometry experiments. Below, we provide below the biomass results for the reviewer's interest. However, we do not believe that these estimates should be compared among treatments, since the biomass on day 8 can differ among populations due to small difference in inoculation density or growth rate, rather than metabolic efficiency.

R1: It is also unclear to me that it is possible to rule out changes in nutrient uptake given the data. While I buy that all treatments are altering expression of proteins associated with carbon metabolism, it is not clear that they are not also altering uptake kinetics. That being said it is interesting that parallel changes in protein expression are observed in the face of different stresses.

We had also performed nutrient uptake experiments along with the stoichiometry experiments, and now provide these results below for the reviewer's interest. The experiments were performed by quantifying the dissolved nitrate and phosphate remaining in the medium after 8 days of culturing.

We observe that in the Control treatment, a greater concentration of the original dissolved nitrate (NO_3^-) is remaining in the medium at the end of the experiment. We also see that there was a greater concentration of dissolved phosphate (PO_4^{3-}) at the end of the experiment in the Control, Low-Phosphorus and Biotic treatments. This indicates that in these treatments, uptake rates per unit biomass decreased significantly. However, since this effect is also observed in the Control treatment, it may simply be due to adaptation to growth in the chemostat environment, rather than a response to any of the experimental selection treatments *per se*. Since we were interested in phenotypes associated with response to selection under low resources (and osmotic stress) and not to life in chemostat, we did not include these results in manuscript. However, if the reviewer feels that this is an interesting result, we can incorporate it into the Supplementary Information.

R1: Figure 1 - On my screen (and printer) the control and salt treatment are both red. It would be useful to label each treatment a unique color.

We have adjusted the figure accordingly.

Reviewer 2:

R2: The new measurements provide information about the stoichiometry of biomass, indicating that the selection lines have optimized their metabolism for efficient carbon fixation relative other cellular components. Here, still the question remains how this could be explained, whether this is due to changes of photosynthetic capacity and CO₂ uptake or metabolic fluxes. Although her authors argue that the focus of the paper is on the proteomic analyses, these data are required for a journal with broader impact. The measurements of photosynthetic capacity and CO₂ uptake should be straightforward.

In response to the reviewer's request, we performed photosynthetic carbon assimilation and respiration measurements and now present the results in Figure S09. Respiration measurements were performed by measuring the rate of oxygen depletion over a dark incubation period of one hour. Carbon assimilation was measured by determining the rate of in situ ¹⁴C assimilation. These results have been standardized per unit chlorophyll (chlorophyll-a, -b and lutein).

In line with our expectation based on the protein expression results, we observe a significant decrease in respiration in the Low-Light, Low-Nitrogen, Low-Phosphorus and Biotic selection treatments. Many of the proteins in the mitochondria are down-regulated in the selection treatments at the same time that rates of respiration declined.

Contrary to our expectation however, we also observe a decrease in carbon assimilation in the Low-Nitrogen and biotically-depleted medium selection treatments. This is despite the fact that the majority of proteins in the chloroplast were up-regulated. This suggests that protein up-regulation in the chloroplast occurred despite simply maintaining the same, or even lower, levels of photosynthesis relative to that observed in the ancestors. This appears to be an example of phenotypic buffering, where evolutionary changes at the molecular level occur while functional changes at higher levels of biological organization "stand still". These evolved molecular phenotypic changes may have evolved to buffer the long-term negative impacts of the low resource supplies and osmotic stress on photosynthesis.

We generally did not observe a significant change in the ratio of photosynthesis to respiration in the selection treatments, except in the Low-Phosphate selection treatment where the photosynthesis to respiration ratio increased significantly relative to that in the ancestors. This means that carbon metabolism only became more efficient in terms of carbon fixed per carbon respired in the Low-Phosphate treatment. This suggests that although generally less carbon is respired after long-term low-resource selection, there is not a generally significant increase in the net rate of carbon fixation.

Together with the stoichiometry data, these results lead us to conclude that while carbon metabolism itself did not generally become more efficient, the amount of carbon fixed relative to the assimilation of other limiting resources (i.e. nitrogen and phosphorus) increased (Figure S08). We discuss this on lines 308-332 of the manuscript.

R2: In regard to the mutated peptides, the authors used MaxQuant and found 1,142 mutated peptides out of 1,417,605 non-mutated one. Do these mutations cluster certain functional entities? Could the mutations explain phenotypic differences?

The mutations are distributed randomly across the strains ($p = 1$, chi-square test) and therefore are unlikely to explain any phenotypic differences. Furthermore, an enrichment analysis of the mutated peptides shows a distribution of functions which closely resembles the functions listed in Figure 3: photosynthesis, ribosomal proteins, carbon metabolism, ATP binding etc.

REVIEWERS' COMMENTS:

Reviewer #1 (Remarks to the Author):

This manuscript provides data on differences in proteome between evolved lines of an alga. The data on patterns of proteome change may be useful to the field. I find the arguments for the selective forces driving the patterns less convincing however sufficient detail is provided for readers to evaluate the science presented.

I would encourage the authors to include the biomass, and nutrient/biomass data in a supplemental figure. While the pattern for nitrogen appears mostly driven by differences in biomass, there may be more divergence in the phosphorus uptake.

Reviewer #2 (Remarks to the Author):

The manuscript has clearly improved with the new data.

The authors still argue that cells use increased metabolic efficiency as an adaptive strategy to non-substitutable resource limitation, despite the fact that no increase in carbon assimilation was observed. Although latter, is discussed, the increased metabolic efficiency argument is unclear. This should be revisited.

They state while carbon metabolism did not generally become more efficient under long-term low-resource selection, C:N and C:P biomass ratios tended to increase, indicating that nitrogen and phosphorus assimilation efficiency improved. Here I agree and suggest to link this to what it mentioned as "increased metabolic efficiency".

REVIEWERS' COMMENTS:

Reviewer #1:

R1: I would encourage the authors to include the biomass, and nutrient/biomass data in a supplemental figure. While the pattern for nitrogen appears mostly driven by differences in biomass, there may be more divergence in the phosphorus uptake.

We have included the biomass and nutrient/biomass data as Supplementary Figure 7, and refer to this on lines 307-309 of the document manuscript_changes.docx which includes the tracked changes, and corresponds to lines 231-235 of the document manuscript.docx which does not include the tracked changes.

Reviewer #2:

R2: The authors still argue that cells use increased metabolic efficiency as an adaptive strategy to non-substitutable resource limitation, despite the fact that no increase in carbon assimilation was observed. Although latter, is discussed, the increased metabolic efficiency argument is unclear. This should be revisited. They state while carbon metabolism did not generally become more efficient under long-term low-resource selection, C:N and C:P biomass ratios tended to increase, indicating that nitrogen and phosphorus assimilation efficiency improved. Here I agree and suggest to link this to what it mentioned as "increased metabolic efficiency".

We have emphasized that by metabolic efficiency, we are referring to nitrogen and phosphorus assimilation efficiencies. We have clarified this on lines 37-39, 135-138, 304-307, 360-362 and 402-403 of the document manuscript_changes.docx which includes the tracked changes, and correspond to lines 30-32, 115-118, 225-228, 264-265 and 298-301 of the document manuscript.docx which does not include the tracked changes.